# KGMistral: Towards Boosting the Performance of Large Language Models for Question Answering with Knowledge Graph Integration

Mingze Li[1], Haoran Yang[1], Zhaotai Liu[1], Mirza Mohtashim Alam[1,2], Ebrahim Norouzi[1,2], Harald Sack[1,2] and Genet Asefa Gesese[1,2]

[1]*Karlsruhe Institute of Technology, Institute AIFB, Germany*

[2]*FIZ Karlsruhe – Leibniz Institute for Information Infrastructure, Germany*

### Abstract

In this paper, a novel question-answering (QA) approach named KGMistral is proposed, based on the Retrieval Augmented Generation (RAG) framework. Given the limitations of Large Language Models (LLMs) in generating accurate answers for domains not adequately covered by their training corpus, this work focuses on leveraging external domain-specific Knowledge Graphs (KGs) to enhance the performance of LLMs. Specifically, the study examines the benefits of using information from a KG to improve the QA performance of the Mistral model in the material science and engineering field. Experimental results indicate that KGMistral significantly enhances Mistral's QA performance.

### Keywords

Knowledge Graph, Large Language Models, Question Answering, Retrieval Augmented Generation, SPARQL

## 1. Introduction

Large language models (LLMs) pre-trained on large-scale corpora, have demonstrated powerful capabilities in revolutionizing various natural language processing (NLP) tasks in different domains such as law [1], medicine [2], and education [3]. One major area being revolutionized by LLMs is question answering (QA), as these models enable users to ask questions and retrieve answers in natural language [4]. However, LLMs have faced criticism for generating incorrect answers, known as hallucinations, which pose a significant challenge to their reliability [5]. For example, LLMs could generate inaccurate medical diagnoses or treatment recommendations, leading to potentially catastrophic risks [6].

To mitigate these issues, one potential solution is to integrate information from external sources such as domain specific KGs into LLMs. KGs store vast amounts of information in the form of triples, comprising a head entity, a relation, and a tail entity, offering a structured and comprehensive knowledge representation [7]. Domain-specific KGs tailored to particular fields can provide accurate and reliable information, such as the MSE KG [8] in the materials science and engineering domain.

This work aims to answer the following research question:

- *How to improve the QA performance of LLMs, specifically Mistral, by integrating a domain-specific KG?*

The key contributions of this work are given as follows:

- *A novel architecture named KGMistral, based on the Retrieval Augmented Generation (RAG) framework, is proposed.*

*DL4KG'24: Workshop on Deep Learning and Large Language Models for Knowledge Graphs, ACM KDD'24, August 26, 2024, Barcelona, Spain*

✉ uemxh@student.kit.edu (M. Li); ubgjw@student.kit.edu (H. Yang); uvyrq@student.kit.edu (Z. Liu); Mirza-Mohtashim.Alam@fiz-karlsruhe.de (M. M. Alam); Ebrahim.Norouzi@fiz-karlsruhe.de (E. Norouzi); Harald.Sack@fiz-karlsruhe.de (H. Sack); Genet-Asefa.Gesese@fiz-karlsruhe.de (G. A. Gesese)

🌐 https://www.fiz-karlsruhe.de/en/forschung/lebenslauf-prof-dr-harald-sack (H. Sack); https://www.fiz-karlsruhe.de/en/forschung/lebenslauf-und-publikationen-dr-ing-genet-asefa-gesese (G. A. Gesese)

🆔 0000-0001-7069-9804 (H. Sack); 0000-0003-3807-7145 (G. A. Gesese)

- *The use of SPARQL queries to retrieve relevant triples (i.e., context) is examined.*

- *A set of experiments is conducted to evaluate KGMistral using competency questions and a KG in the domain of materials science. The results indicate that using domain-specific KGs as external sources in RAG leads to improved QA performance.*

To the best of our knowledge, this is the first work leveraging the RAG framework in the material science and engineering domain by integrating domain-specific KG.

## 2. Background and Related Work

### 2.1. Background

**Mistral 7B**    Mistral-7B [9, 10] is a cutting-edge LLM with 7 billion parameters, designed for high performance and efficiency. It surpasses the top open-source 13B model (LLaMA-2-13B [11]) in all evaluated benchmarks and exceeds the best open-source 34B model (LLaMA-34B [12]) in reasoning, mathematics, and code generation. Mistral-7B leverages grouped-query attention for faster inference and sliding window attention to manage sequences of any length more effectively, all while reducing inference costs as discussed in detail in [13].

**Retrieval Augmented Generation (RAG)**    RAG [14] is a language generation method that improves the accuracy and reliability of generative AI models by incorporating facts retrieved from external sources. This method is built on a tripartite foundation comprising retrieval, generation, and augmentation techniques. By utilizing external knowledge, RAG significantly mitigates the issue of hallucination in LLMs, leading to its widespread adoption [15].

**SPARQL**    SPARQL [16] is a query language for the Resource Description Framework (RDF) data. As a query language, it can be used to add, remove, and retrieve data.

### 2.2. Related Work

To address hallucinations in LLMs, one emerging solution involves using external knowledge as supplementary information to assist LLMs in generating authoritative outputs [17]. In [18], LLMs are trained to retrieve relevant knowledge from external KGs to tackle domain-specific questions. The Mistral model with Contextual Position Encoding (CPE) introduced in [19] dynamically encodes positional data based on token context, thereby improving evaluation outcomes. The issue of hallucinations in Mistral is also addressed in [20] by using RAG to integrate information from Wikipedia, enhancing the model's accuracy. In a related but distinct study [21], an ontology serves as an external knowledge source for developing a Text-to-SPARQL system. In contrast to all these methods, the approach proposed in this paper employs SPARQL queries for retrieving relevant triples from the KG. These triples are then integrated into the prompt for the QA task.

## 3. KGMistral

The general architecture of the proposed approach is given in Figure 1. The various components of the architecture i.e., entity and relation extraction, similarity matching, extraction of relevant triples using SPARQL, verbalization, and prompt engineering and response generation, are discussed in detail in the subsequent sections.

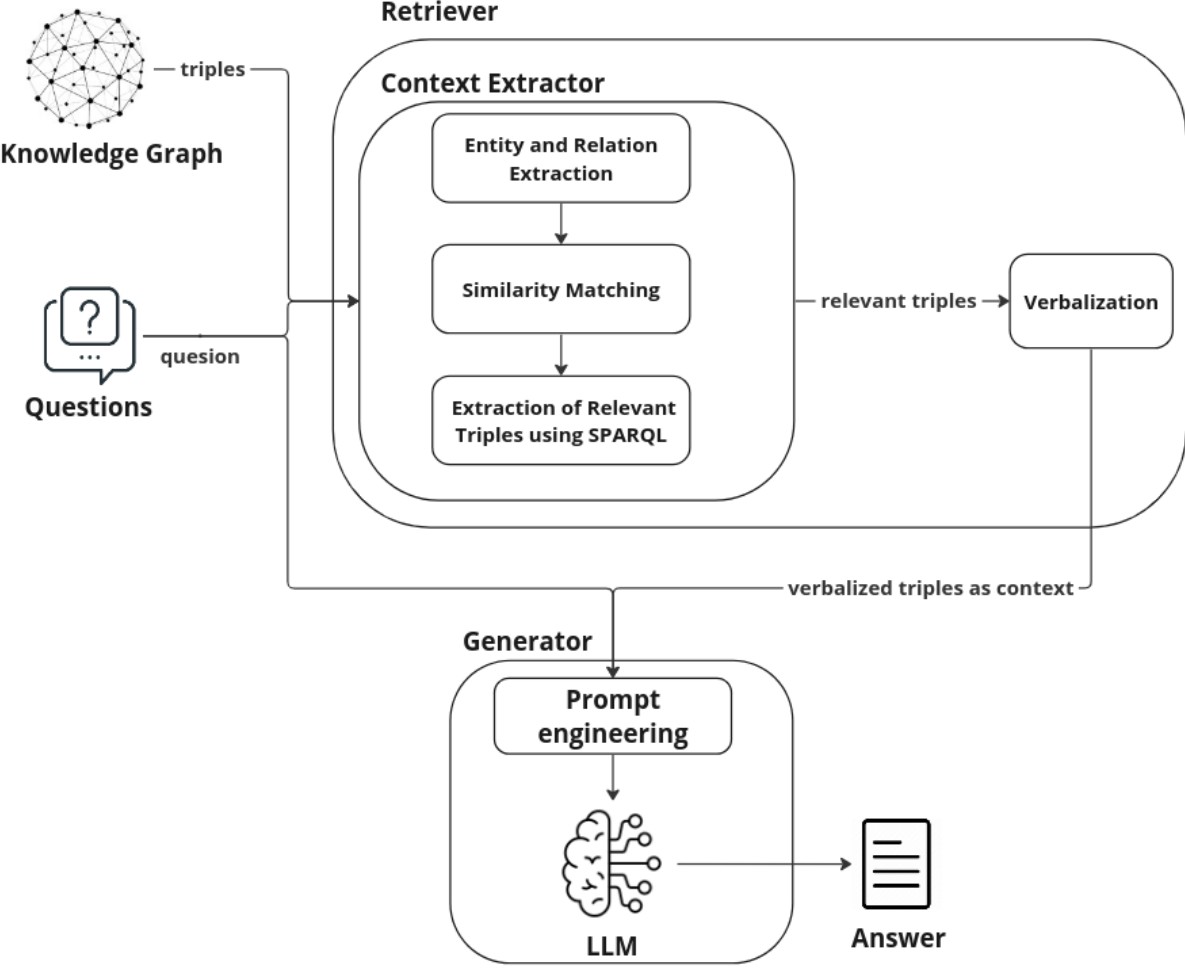

**Figure 1:** The general architecture of the proposed approach

## 3.1. Entity and Relation Extraction

The purpose of this component is to extract entities and relations from the set of questions. The first step in the extraction process involves applying the en_core_web_sm named entity recognition (NER) model from spaCy[1] due to its efficiency and customizability. Relations often exhibit varied expressions in natural language questions, as highlighted by Berant et al. [22]. For instance, the predicate *"email address"* in the question: *"What is the email address of 'ParaView'?"* could be expressed in multiple ways, such as *"What is...'s email address?"*, *"What is... contact point"*, or *"How could... contact...?"*.

To address this challenge, specific regular expressions and part-of-speech analysis are employed to improve the categorization of entities and relations, regardless of their varied expressions. This approach can also enable handling a broad range of questions effectively.

## 3.2. Similarity Matching

After extracting named entities and relations from a given question (see Section 3.1), the next step involves matching them to elements in the KG. This process aims to identify entities and relations in the KG that are relevant to the question. To achieve this, semantic similarity matching is performed for entities using BERT with cosine similarity metric, leveraging the entities' labels and textual descriptions. Similarly, for relations, the Spacy's en_core_web_lg [2] model is used. For each question, the top n

---

[1]https://spacy.io/models/en#en_core_web_sm
[2]https://spacy.io/models/en#en_core_web_lg

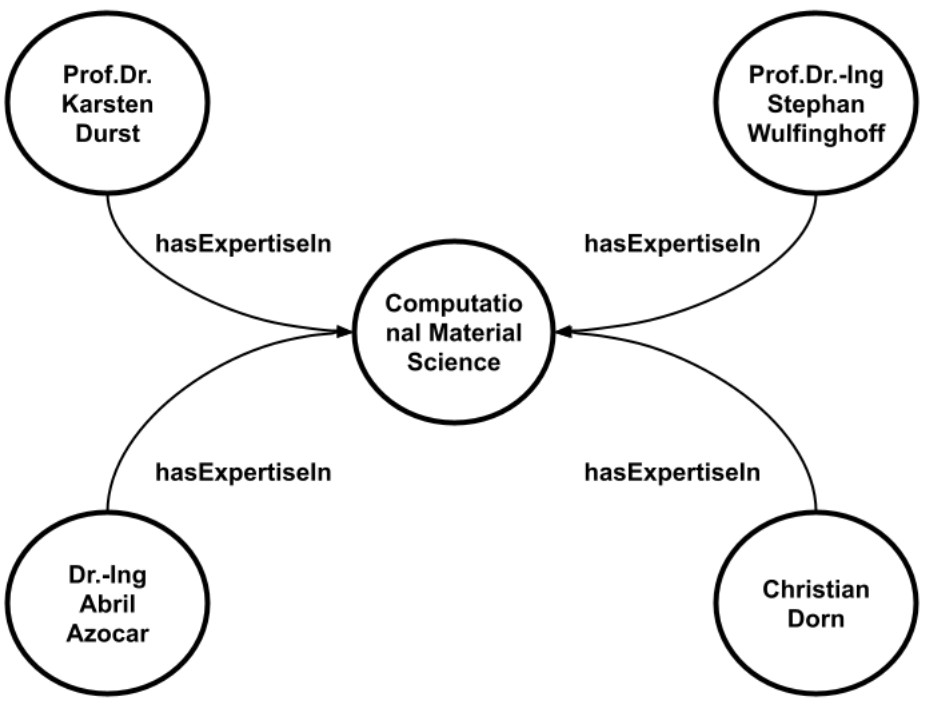

**Figure 2:** A simple KG where entities and relations are depicted in circles and lines, respectively

most similar entities and the top m most similar relations in the KG are identified. In the next step, the identified entities and relations are then utilized to construct SPARQL queries to search for relevant triples from the KG.

### 3.3. Extraction of relevant triples using SPARQL

While some questions are simple, others can be more complex requiring more than one-hop graph traversal. For example, considering Figure 2, the question *"Who is working in the Computational Materials Science field?"* would be a simple query that can be answered by looking at the one-hop neighbors of the entity *"Computational Materials Science"*. On the other hand, the question *"Who is working in the same field as 'Prof.Dr. Karsten Durst'?"*, may require two-hope graph traversal to retrieve the correct answers.

SPARQL can be utilized to answer straightforward questions involving a single entity and a single relation, as well as to infer more complex multi-hop facts. In this work, once an entity and a relation are identified during the similarity matching phase as discussed in the previous section (see Section3.2), a SPARQL query is constructed using the templates provided in Table 1. In this table, SPARQL Template 1 provides the query that can be used to extract entities that appear in the head position in triples where the relation and the tail entities are fixed to some given URIs. On the other hand, SPARQL Template 2 is designed to extract entities that appear at the tail position given some head entity and relation. SPARQL Template 3 is created to extract multi-hop facts. Specifically, it is used to extract more relevant information by first taking a tail entity that is returned as part of the results of a query with Template 2, together with a relation, it retrieves new head entities that are different from the original head entity.

Table 1: SPARQL Templates

**SPARQL Template 1**

```
SELECT ?headEntity
WHERE {
   ?headEntity <relation_uri> <tail_entity_uri> .
}
```

**SPARQL Template 2**

```
SELECT ?tailEntity
WHERE {
   <head_entity_uri> <relation_uri> ?tailEntity .
}
```

**SPARQL Template 3**

```
SELECT ?otherHeadEntity ?tailEntity
    WHERE {{
        {{
            SELECT ?tailEntity
            WHERE {{
                <head_entity_uri> <relation_uri> ?tailEntity .
            }}
        }}
        ?otherHeadEntity <relation_uri> ?tailEntity .
        FILTER (?otherHeadEntity != <head_entity_uri>)
    }}
```

The answers that are retrieved using the SPARQL queries will then be passed to the verbalization phase where they will be processed and converted to sentences that make proper sense (see Section 3.4).

## 3.4. Verbalization

Since the results returned by SPARQL are URIs, it is necessary to verbalize them so that LLMs would be able to make sense of them. Verbalization is performed by replacing the entities and the predicates in the triples with their corresponding human-readable labels.

## 3.5. Prompt Engineering and Response Generation

The verbalized triples created using the previous steps are used as context and passed to the prompt engineering and response generation step to generate answers from LLMs Specifically Mistral 7B (without loss of generality) as illustrated in Figure 1. The prompt consists of three components: instruction, relevant information for enhanced context (verbalized triples), and the user question. The structured prompt is delineated as follows:

**System Instruction:**

- **Role and Purpose:** "You are a helpful assistant. Extract and answer using key information from context." This instruction sets a clear expectation for the system's function, explaining the idea of

boosting LLM performance with relevant contextual information verbalized from triples.

- **Precision and Brevity:** "Ensure the response is concise, without duplicates, focusing solely on crucial details." This directs the LLM to avoid redundancy and extract only essential details.

- **Examples:** Two examples were provided to clarify the expected response format:
  - Example 1: (Context: The sun is a star in the center of our solar system. Question: What is the sun? Answer: A star at the center of the solar system.)
  - Example 2: (Context: Neil Armstrong was the first person to walk on the moon. Question: Who was the first person to walk on the moon? Answer: Neil Armstrong.)

- **Instructions on Format:** "Your answer must be provided in a direct and concise format, without using any lead-in format such as 'Answer:' or similar. Only the answer itself should be included in the response." This instruction ensures the LLM generates a simplified, correct format response, increasing evaluation performance.

**Relevant Information for Enhanced Context:**

- This component utilizes the verbalized relevant triples from the KG as context for each question.

**User Question:**

- The user question is also part of the prompt to be fed into the LLM along with the instruction and context to generate responses, which is a key step in RAG.

Using this prompt, the proposed architecture efficiently integrates the relevant triples from a KG into LLMs for improved QA.

## 4. Experiments

In this section, the experiments conducted to evaluate the performance of the proposed approach are presented. The source code and the datasets are made publicly available at https://github.com/Mingze101/KGMistral.

### 4.1. Dataset

The MSE-KG[3], representing data from institutions within the NFDI-MatWerk consortium[4], is used as an external resource for retrieving relevant triples. The KG contains information on (i) relevant community structure: researchers, research projects, universities, and institutions; (ii) infrastructure: software, workflows, controlled vocabularies, instruments, facilities, educational resources, and events; and (iii) data: repositories, databases, scientific publications, published datasets, and reference data. MSE-KG is composed of 8,166 triples, 112 relations, and 1823 entities. The number of competency questions used for the experiments is 37.

### 4.2. Baselines

KGMistral is compared against three baselines:

- **Mistral**: This model operates without utilizing any information from the KG, relying exclusively on the knowledge contained within the Mistral LLM.

---

[3]https://demo.fiz-karlsruhe.de/matwerk/
[4]https://nfdi-matwerk.de/

- **Mistral$_{\text{Raw}}$**: In this model, the input KG is divided into chunks using a character-based text splitter, with each chunk having a maximum length of 1024 characters. Next, vector similarity is calculated between the question text and these chunks using a pretrained LLM. The top K chunks with the highest similarity scores (nearest neighbors) to the question are then selected as the context. Note that this model does not utilize SPARQL.

- **Mistral$_{\text{Verbalized}}$**: This baseline model is a verbalized version of Mistral$_{\text{Raw}}$, where each triple from the KG is converted into a sequence resembling a natural language sentence, making it easier for the LLM to understand.

### 4.3. Experiment settings and Evaluation Metrics

The hyper-parameters for similarity matching, n and m, are set to 5 and 9 respectively. To prevent the Mistral 7B model [10] from becoming overly creative and deviating from the answer, its perplexity is set to zero. The metrics given in Table 2 are used to evaluate the models. BLEU[23] measures how many words and phrases in the machine-translated text appear in the reference translations, taking into account the order of words through the use of n-grams. ROUGE[24] assesses the quality of a generated summary or translation by comparing it with one or more reference texts.

**Table 2**
Evaluation Metrics

| Method | Formula |
|---|---|
| BLEU | $\text{BLEU} = BP \cdot \exp\left(\sum_{n=1}^{N} w_n \log p_n\right)$ |
| Rouge | $\text{ROUGE} = \sum (\text{Recall of } n\text{-grams})$ |

### 4.4. Results

As shown in Table 3, the models **Mistral$_{\text{raw}}$**, **Mistral$_{\text{verbalized}}$**, and **KGMistral** which utilize the KG as an external information source, outperform the **Mistral** model that does not leverage the KG. This demonstrates that integrating KGs into QA systems using RAG leads to significant improvements. The results also reveal that verbalization enhances model performance by making the retrieval process more effective. Notably, the proposed **KGMistral** approach surpasses all baseline models w.r.t all metrics except BLEU, indicating that generating context for user questions by extracting relevant triples with SPARQL queries and then verbalizing these triples is highly promising.

### 4.5. Limitations

In this work, the relation and entity extraction component of the proposed approach is primarily designed for simple user questions that include only one entity and one relation. Consequently, applying it to more complex questions may result in degraded performance.

## 5. Conclusion and Future Work

In this study, the advantages of integrating information from KG to enhance the QA performance of LLMs in the field of materials science and engineering are investigated. A novel RAG-based QA approach named KGMistral is proposed. According to the experimental results, KGMistral outperforms all the baseline models. Despite these promising results, further improvements can still be made. Therefore, the following directions can be investigated in future work:

- Improving the relation and entity extraction process to support more complex user question

- Enhancing the verbalization process

- Experimenting with other LLMs, such as GPT-3.5-turbo and fine-tuned GPT-3.5-turbo.

**Table 3**
Evaluation Results with the best values written in bold

|  | BLEU |  | Rogue-1 | Rogue-2 | Rogue-L |
|---|---|---|---|---|---|
|  |  | **$F_1$** | 0.04 | 0.019 | 0.006 |
| **Mistral** | 0 | **Precision** | 0.033 | 0.018 | 0.036 |
|  |  | **Recall** | 0.077 | 0.007 | 0.030 |
|  |  | **$F_1$** | 0.078 | 0.037 | 0.085 |
| **Mistral$_{raw}$** | 0.025 | **Precision** | 0.107 | 0.060 | 0.106 |
|  |  | **Recall** | 0.126 | 0.043 | 0.131 |
|  |  | **$F_1$** | 0.108 | 0.053 | 0.115 |
| **Mistral$_{verbalized}$** | 0.018 | **Precision** | 0.138 | 0.077 | 0.135 |
|  |  | **Recall** | 0.144 | 0.070 | 0.148 |
|  |  | **$F_1$** | **0.150** | **0.071** | **0.134** |
| **KGMistral** | 0.008 | **Precision** | **0.156** | **0.079** | **0.143** |
|  |  | **Recall** | **0.245** | **0.120** | **0.225** |

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
