# OpenReview forum: "KGMistral: Towards Boosting the Performance of Large Language Models for Question Answering with Knowledge Graph Integration"
_KDD.org/2024/Workshop/DL4KG — DL4KG 2024_

### Official Review · Reviewer_knBq · 2024-07-03
**The authors introduce the usage of SPARQL for integrating KG facts into LLM prompts for Question Answering to mitigate Hallucinations.**

**Rating:** 6
**Confidence:** 3

**Review:**

The authors describe a method of integrating external Knowledge from KGs in LLM prompts for improved QA performance. Their approach consists of extracting relevant triples from the KG and verbalizing them to add as context to an LLM prompt. The process from KGs to verbalized context is done in three steps. First, the entities and relations from the questions are extracted via pre-trained models. In the next step, the extracted entities and relations are matched with their corresponding KG entries. Based on this, SPARQL queries are built via templates to extract the relevant triples from the KG. The verbalization process consists of replacing the entities and predicates from the triples by their corresponding labels in the KG.
In the prompting step, the extracted and verbalized triple is added as context to the question for the LLM prompt. The main novel contribution is the usage of SPARQL for retrieving the triples from the KG. The authors show that their three step approach performs better than approaches where individual steps are left out.
Overall, the presented work is a case-study and showcases a certain way of integrating KG context into LLM prompts for QA.
Pros:
-	The proposed usage of SPARQL in the process of context generation leads to an improved performance in the Rouge Scores.
Cons:
-	Presentation quality could be improved (i.e. Fig. 2 appears to be low-res, Table 1 is cut-off at a newpage)
-	Evaluation is done only on one KG in one domain
The topic of hallucination is very important and the paper introduces a way of mitigating this issue, although only in one specific domain. Given the significance of the topic, I tend to accept this paper.

---

### Official Review · Reviewer_foFn · 2024-07-04

**Rating:** 7
**Confidence:** 4

**Review:**

The authors introduce KGMistral, an architecture for Retrieval Augmented Generation (RAG) using the Mistral language model. However, the focus on Mistral seems unnecessary, as the architecture could be adapted for other LLMs.

The paper's format should be CEURART, not LNCS.

The paper is well-written, but evaluating the entire architecture solely based on the final LLM output seems unfair, considering the multiple components involved. An ablation study would help assess the contribution of each component to the overall performance.

Overall, the paper is solid, with a clear research question, extensive evaluation set, and standard metrics.

---

### Official Review · Reviewer_Thmr · 2024-07-07
**Relevant paper to the workshop**

**Rating:** 6
**Confidence:** 4

**Review:**

The paper extends Mistral with triples from a KG via RAG and applies the extension in a QA exercise within the domain of Material Sciences.

Comments:
- As the paper focuses on LLMs like Mistral, I was wondering why the steps "Entity and Relation extraction" and "Similarity Matching" do not leverage the power of Mistral. This seems to affect performance in more complex queries as highlighted in Section 4,5
- About the verbalization: what about the cases where the entities may have more than one label/synonym?
- Mistral_raw: using triples from KG but without SPARQL --> ??  What is the difference here with respect to KGMistral. What is the role of the (results of the) SPARQL queries for KGMistral?
- Scores seem to be low. Moe LLMs should be evaluated in the future and possibly different ways of performing the entity extraction and the similarity matching to understand the impact of the various choices.
- I do not think the RAG-based approach is especially novel, but worth exploring different architectures and application domains.

The following paper is relevant as the authors extract SPARQL directly from a natural language description and an ontology:
- Dean Allemang, Juan Sequeda: Increasing the LLM Accuracy for Question Answering: Ontologies to the Rescue! CoRR abs/2405.11706 (2024)

---

### Decision · Program_Chairs · 2024-07-09

Accept